ͨ | **Editor's Pick** | Conduct of Scientific Research | Minireview

# Race by other names: critiquing population categories in microbiome research

**Marta Scaglioni**[1]

**ABSTRACT**   This minireview analyzes the social science literature on how categories of human difference are conceptualized and deployed in microbiome science, particularly in research conducted in Global South contexts. Through a systematic review of 180 microbiome studies (2014–2025) and analysis of social science critiques, it demonstrates how seemingly neutral terms like "population," "lifestyle," and "traditional" often imply racial meanings that echo colonial scientific frameworks. While microbiome science rarely uses explicit racial terminology, it frequently employs comparative frameworks that create hierarchical distinctions between Western and non-Western populations. The minireview argues that these categorizations stem not just from colonial legacies but from the comparative nature of scientific knowledge production itself. It traces the evolution of racial thinking in science from post-World War II genetics through contemporary microbiome research, showing how even as scientists reject biological racial determinism, race persists as an analytical category through various proxies. It concludes by examining recent efforts to develop more nuanced approaches to human categorization in microbiome science and calls for deeper integration of social science perspectives to avoid reproducing problematic categorizations while ensuring ethical and equitable research practices.

**KEYWORDS**   microbiome science, race, populations, Global South, scientific categorization, comparisons

## RACE AND POPULATION CATEGORIES IN SCIENTIFIC PRACTICE

Behind the façade of scientific objectivity, race continues to haunt modern science. Contemporary biomedical research often employs categories of human difference that have recently come under criticism by the U.S. National Institutes of Health for having generated more confusion and harm than precision or accurate disease prediction (1). Race remains what it has always been: a socio-political construct that divides humans based on appearance and presumed ancestry yet is frequently misused in scientific research as a surrogate for genetic variation between populations.

While almost all scientists agree that biological race—the notion that our species is divided into genetically discrete categories—is scientifically inconsistent, debates persist about race's utility as an analytical category in research. Three distinct perspectives have emerged: first, mostly geneticists argue that racial categories obscure rather than illuminate, emphasizing that human genetic variation does not map onto socially defined racial groups (2) (including the aforementioned commissioned study "Using population descriptors in genetics and genomics research" [1]). Second, epigenetic researchers often retain race as an analytical tool, using it as a proxy for socioeconomic and demographic factors including healthcare access, financial stability, and social stress (3). In this latter framework, race becomes operational—not as a biological reality, but as an organizing principle to understand how social inequities become embodied and manifest in health disparities (4). A third position is held by clinicians who strongly

**Peer Reviewers** Joseph L. Graves, North Carolina Agricultural and Technical State University, Greensboro, North Carolina, USA; Abigail Nieves Delgado, Utrecht University, Utrecht, the Netherlands; Lianmin Chen, University of Groningen, Groningen, the Netherlands

Address correspondence to Marta Scaglioni, scaglionimarta@gmail.com.

The author declares no conflict of interest.

See the funding table on p. 7.

believe in the importance of social race to monitor, understand, and intervene on differences in health, but who reject the term and advocate for a replacement with "ethnicity" or "ethnic group," considered to be less biology-based (5).

Making the case specifically through microbiome research, this minireview aims at making the point that categories such as "ethnicity," "ethnic group," "population," "lifestyle," and "communities" hide racial subtexts, especially when they are used to describe Global South contexts (the notion of Global South is far from non-controversial, and many scholars replace it with terms such as "neo-colonial"; however, the language of science and of international funding schemes actively employs it) (6). It will focus specifically on microbiome research conducted in Global South contexts, arguing that the categorization observed in microbiome science is highly asymmetrical as the Global North tends to define the Global South and not vice versa. Then, the article connects this dynamic to the inherently comparative nature of scientific knowledge production. Categories used to define human groups in microbiome science, as epistemic tools shaped by specific historical and cultural contexts, reflect not just the mindset of individual scientists, but entire genealogies of Western scientific thought about human difference: colonial scientific practices, post-World War II (WWII) population genetics, and contemporary biomedical frameworks. Their historical embeddedness in Western thought makes them particularly problematic when applied to Global South contexts, where they often fail to capture local social organizations, cultural practices, and demographic realities.

In the first section, I briefly trace the evolution of race categories in scientific research, from genetics to contemporary microbiome science. The second section analyzes how social scientists have critiqued racial categorizations in microbiome studies specifically. Finally, I examine how microbiome research in the Global South employs seemingly neutral terms within comparative frameworks. I conclude by tracing the social and life science literature that offers programmatic suggestions for a more precise, equitable science.

## METHODOLOGICAL APPROACH TO LITERATURE REVIEW

I will delineate the debates around the use of racial categories in genetic and medical research through secondary sources, both social and life science studies. The paragraph "From population to lifestyle: analyzing category usage in microbiome papers," instead, will be based on an online search on PubMed and SCOPUS where I collected 180 papers from 2014 to 2025 that deal specifically with human microbiome and the Global South (mainly searched through the keywords "developing countries"). This literature review was made possible through institutional access to academic journals via university subscriptions, which themselves represent a significant structural asymmetry in knowledge production. The ability to comprehensively review Global South micro-biome research is primarily available to Global North researchers, and Northern scholars can more easily study and write about Southern contexts than vice versa.

## A BRIEF GENEALOGY OF RACE IN SCIENCE: FROM WWII TO EPIGENETICS

### Post-war rejection of racial determinism

While this brief overview starts from WWII as a turning point, racial categorization has a long history in biology, from Linnean taxonomy to 19th century scientific racism (7, 8) . These classifications often served to justify colonial exploitation and social hierarchies through appeals to scientific authority (9). The post-war rejection of racial determin-ism thus represented a significant break from deeply entrenched practices of racial categorization in Western scientific thought. WWII, as a watershed, marked a consistent decrease in the use of categories of race in science, supported by the 1950 UNESCO statement on race. After WWII, U.S.-based epidemiological and medical research relied little on explicit racial categories, often replacing them with geographical and social coordinates (10).

With the advent of the genetic study of human difference, geneticists' progressive agenda was to confute the biological consistency of race through nuclear and mitochondrial DNA analysis and later through the analysis of the Y chromosome (2). Several genetic studies successfully disproved the biological distinction between groups, showing how intra-population diversity is higher than inter-population diversity (11–15). The relatively young evolutionary age of our species, *Homo sapiens* (approximately 300,000 years), combined with persistent patterns of human migration, has worked against substantial genetic differentiation between human groups (16).

In the 1990s, the Human Genome Project (HGP) and the Human Genome Diversity Project (HGDP), launched by the U.S. government, were the first comprehensive projects that aimed to map the entire human genome using computational techniques. As social sciences have shown, the HGP reflected a color-blind logic—attempting to create a "universal human model" (17) —but relied heavily on genetic samples already collected in Europe (18). HGDP emerged as a response to criticism of HGP being "ethnocentric" (19, 20) and introduced other racial and ethnic categories within a broader shift towards including racial minorities (18). The subsequent successful sequencing of the human genome in the 2000s promised the end of race as a proxy for genetic difference (21). These post-racial promises did not last long (19, 22, 23): HGDP was harshly criticized by physical anthropologists (22, 24) and minority groups (e.g., Indigenous Peoples Council on Biocolonialism [see http://www.ipcb.org/]) for, among many other ethical issues, essentializing racial categories and relying on broad misinformed population categories.

As social scientists have shown, since the 1990s, geneticists started replacing the category of race with "population" or "group" (19, 25), that, however, are conceptually kindred to race (25). These studies revealed that these categories' meaning and significance are always produced through social interventions and reflect the perspective of the scientists using them. Social science studies noted the tendency to uncritically overlap population categories with geographical origins (10), revealing how these categorizations were based on a misinformed idea of populations, especially regarding those residing outside the West. For example, the International HapMap Project, launched in 2002 to map human genetic variations across different populations worldwide, generated criticism because the sampled categories resembled continental categories (African, European) (26) and thus appeared oversimplified.

In short, notwithstanding the attempt to dismiss the scientific consistency of race, these genome projects did not lead to the deconstruction of it. Anthropological inquiry shows how they even entailed the potential to recast race in politically empowering terms (18). This tension between rejecting biological racial determinism while grappling with race's social reality would take on new dimensions with the emergence of epigenetics.

## Beyond genes: race and the epigenetic paradigm

The epigenetic paradigm marks a shift from a gene-centric view to one that examines how environmental factors influence gene expression. Unlike human genome sequencing efforts, which relied on place-neutral medicine and a clear organism-environment distinction (27, 28), epigenetics emphasizes how local contexts and environments shape biology. This paradigm shift coincided with a resurgence of race categories in biomedical research, though in a different form (10, 26). Epigenetic studies employ it as an analytical variable to track health disparities and environmental influences. Life scientists King et al. (3) define race in epigenetic research as a "variable of interest," or as a "proxy for socioeconomic and sociodemographic variables, including financial stability, healthcare access, and social stress, among others." (3). This attention to race as an analytical category emerged primarily in U.S.-based research, where racial health disparities have been a central focus of biomedical investigation (10).

In microbiome science, researchers have demonstrated strong correlations between microbial diversity and social factors that are often racially patterned, such as socioeconomic status, healthcare access, diet, and environmental exposures (29, 30). The Human

Microbiome Project (HMP) (https://hmpdacc.org/), launched in 2007, is a groundbreaking scientific initiative that focuses on studying the genomic composition of all microbes that live on and in the human body, analyzing their collective DNA and investigating their role in human health and disease. It considers racial backgrounds in associations with microbiome diversity and health-related outcomes, concluding that the association is "strong" (31). The racial categories employed within the HMP—termed sometimes "race" and sometimes "ethnicities" —are Asian, Black, Mexican, Puerto Rican, and White, clearly referring to a specific Western genetic research tradition (6). Many microbiome studies in the United States employ these same racial categories (32–35), or refer to African Americans as opposed to people of European ancestry (36), or other nationalities as opposed to White Caucasians (37).

In general, this reintegration of race into biomedical research carries similar risks. Critics from the social sciences warn of a "molecular reinscription" of race (38), which not only carries with it a re-biologization of racial categories (18) but also racist narratives (38, 39). In addition, while epigenetics aims to understand how social inequalities become biologically embodied, its biosocial approach may inadvertently obscure crucial socioeconomic factors affecting health outcomes (40).

This persistence of racial thinking in supposedly post-racial science becomes particularly evident in emerging fields like microbiome research, where social science has recently criticized the uncareful use of race categories.

## CLASSIFYING DIFFERENCE

### Population, ethnicity, and lifestyle in microbiome science

The following sections examine how categories of human difference operate in microbiome science through two complementary perspectives. First, I review how social scientists have analyzed the use of racial thinking in microbiome research. Then, through a systematic analysis of microbiome research papers focused on Global South populations, I demonstrate how these categories are deployed in practice.

### The ghosts of race: social science critiques of microbiome research

As a scientific effort born in the Global North (41, 42), microbiome science moves towards the Global South as the byproduct of specific infrastructures of knowledge and technologies (43). Its first steps in the Global South are primarily sampling endeavors—with the effect of creating and reifying opposing categories: on one side Westerners, presumed to share similar characteristics and lifestyles, and on the other Indigenous, developing countries populations, or non-Westerners, defined by contrast. The second category comprises Black and brown bodies, for whom microbiome studies operate as racialization (6) or "microbiomization" (44, 45) processes. According to STS scholar Andrea Núñez Casal, the categorization of non-Western populations in microbiome research emerges from two distinct but interrelated frameworks. First, an inclusionary framework seeks to address health disparities by incorporating minority groups into research protocols. Second, a bioprospecting framework drives the search for novel microbial diversity in non-Western populations, often leading to the extraction and commodification of biological resources regardless of researchers' conscious intentions (42, 45–49).

Recent social science scholarship has critically examined how microbiome scientists conceptualize non-Western bodies and lifestyles. Scholars have demonstrated how broad categorizations, such as "non-Western," "non-industrialized," "traditional," "non-urbanized," and "rural" serve as imprecise, stereotypic proxies for "under-developed" (50) or "undeveloped" (51) populations. These categories have been termed "ghost variables" (50), as they hiddenly imply a racial subtext. In contrast, Núñez Casal defines race in microbiome research as not just operational or incidentally present, but as foundational to microbiome science' "experimental system," thus challenging its "ghost" nature (45). She delineates these categories as socio-cultural constructions that operate

at different levels: ethnic (the Hadzabe, a Tanzanian hunter-gatherer community), social (industrialized), national (Burkina Faso), or international/political (European) (45). In projects like the HMP, which originate from a Western scientific perspective, classifications of Western populations tend to be broader than those applied to non-Western groups. This asymmetry reflects a dynamic that reinforces Western authority to define "others" while avoiding critical self-examination (45). In general, race categories have come under criticism for being Western-centric, oversimplified, and imprecise (49), leading social sciences to suggest discarding them from microbiome research and focusing on racism (52, 53). Philosophers Abigail Nieves Delgado and Jan Baedke claim that the concept of race used in microbiome science presents a rigid, stable taxonomic nature, while human microbiomes spread and are transmitted in more fluid ways (52).

Specifically, social science analysis has delved into the meanings attached to these categories used to describe "non-Westerners," particularly Africans, highlighting problematic conceptual frameworks. These "histories of otherness" (45) view Global South microbes in different ways. They are conceptualized as "lost microbes"—idealized, highly diverse microorganisms that Western populations supposedly lost through "westernization" and "life in modernity" (48, 52, 54). This discourse by microbiologists reifies "industrialized" populations and lifestyles, attributing noncommunicable and autoimmune diseases to factors like antibiotic overuse, "Western" diet adoption, and reduced environmental microbial exposure. The underlying assumption is that scientists can find more diverse microbiota or "missing" microbes in non-Western populations. The focus on "rural" environments, in fact, implies that microbial diversity follows an urbanization gradient, where practices associated with modernity—such as frequent cleaning, modern architecture, vaccinations, and antibiotic use—are viewed as microbiota-impoverishing (42, 49, 52).

This romanticized construction of rural environments as pristine habitats for "good" microbes has faced substantial criticism (6, 46, 52, 53). Such framing has enabled concerning attempts by Western researchers to appropriate allegedly "missing" microbes from Global South populations (6, 45, 49, 52, 55), as exemplified by microbiologist Jeff Leach's widely publicized fecal transplant involving Hadzabe microbes (56). Also, this perspective problematically frames non-Western populations as "living fossils" (46) or "walking biobanks," (54) collapsing distinctions between non-Western and pre-modern peoples. The assumption of higher microbial diversity in non-Western contexts relies on a simplistic co-evolutionary framework that treats contemporary non-Western individuals as proxies for ancestral humans with "ancient" microbes (45, 57). This framework suggests a straightforward human-microbe co-evolution that has been increasingly questioned within microbiology itself (54). Some critiques have emerged from within affected communities themselves—notably, Hadzabe community member and human rights activist Shani Msafiri Mangola, who has collaborated with microbiologists to challenge the romanticization of indigeneity (see also 58) as representing a lost connection to land and nature (57). Furthermore, microbiome research based on these racial assumptions often displays shallow contextual knowledge (6, 49) and overlooks local microbiota-impoverishing practices (abuse of antibiotics, diffusion of formula milk) and high infection rates in the Global South (54).

Building on these social science critiques of how microbiome research conceptualizes and operationalizes race, a systematic analysis of how scientific papers actually deploy these categories reveals both the pervasiveness of racial thinking and its evolution into seemingly neutral descriptors of human difference.

## From population to lifestyle: analyzing category usage in microbiome papers

A systematic review of 180 human microbiome studies focused on non-Western populations (2014–2025) reveals how comparative frameworks drive the categorization of human groups. As STS and anthropological theorizations of comparison have shown (59–62), the very act of comparing requires and produces boundaries between objects. Through the act of comparison, scientists draw boundaries between

populations, creating seemingly distinct groups that become reified and naturalized through repeated scientific practice. These comparative practices do not simply describe pre-existing population differences—they actively construct and maintain the boundaries between groups, as comparisons can actually shape epistemic perspectives and practices.

Among the analyzed papers, while explicit racial terminology is rare (6.1% of papers), comparative frameworks often generate implicit hierarchies through paired descriptors. Most studies (78.3%) structure their analysis through explicit comparisons, predominantly between "Western"/"non-Western" populations (46.1%), "rural"/"urban" groups (62.5%), or "traditional"/"modernized" societies (26%). The studies predominantly take place in Africa (52.2%), Asia (49%), and Latin America (33%). Within these regions, most studies structure their analysis through explicit comparisons with Western populations (78.3%). In African studies, comparisons primarily involve European (41.5%) or North American (42.2%) reference populations, with a focus on South Africa, Ghana, and Ethiopia. Asian studies most frequently compare populations to North American references (38.4%) or European ones (35.2%), with a concentration in South and Southeast Asia. Latin American research shows similar patterns, with 44.6% using North American and 33.7% using European comparison groups, primarily focusing on Brazil and Peru. These comparative pairs do not just describe differences—they actively construct them through the epistemic practice of comparison itself, unveiling the performative nature of race. As anthropologist Matei Candea (62) argues, such comparisons often operate through what he terms "frontal comparison"—where unfamiliar contexts are measured against a familiar (Western) background of assumptions—rather than through more equitable "lateral comparison" between contexts considered on their own terms. The pervasive use of Western populations and lifestyles as a default comparative reference (appearing in 46.1% of papers) thus reflects not just colonial hierarchies, but the very structure of scientific knowledge production through comparison. Terms like "rural" populations (39.4%), "traditional" lifestyle (32.2%), and references to Indigenous communities (21.1%) gain their meaning primarily through their comparative relationship to an implied Western standard.

Among these studies, many are based in Eastern Africa, with the Hadzabe of Tanzania appearing in 11.7% of all papers—a striking overrepresentation of one community. The focus on such populations has contributed to a shift in microbiome science. Unlike human genetic studies, which often reinforce population categories, microbiome research has increasingly revealed the primacy of lifestyle factors over population-level distinctions. This shift is evident in high-profile studies like Smits et al.'s (63) *Science* cover story. While the microbiological study initially framed its comparison of 18 populations across 16 countries through an "industrialized/traditional" binary, its findings fundamentally challenged these categorical distinctions. The discovery that Hadzabe's microbiome changes dramatically with seasonal dietary shifts suggests that microbial communities respond more to lifestyle practices than to population membership. This finding raises two crucial questions: could people in "industrialized" societies achieve similar microbial diversity simply by adopting different dietary practices, and does the "industrialized/traditional" binary have any real analytical value for microbiome science?

## DISCUSSION

### Beyond population categories: recommendations for future research

Increasingly, in fact, microbiome scientists are questioning their use of categories to differentiate humans, their lifestyle, and the environment in which they live. For example, the study by biologists Sahana Kuthyar and Aspen T. Reese (64) on the variations in microbial exposure at the human-animal interface problematizes dichotomies, such as "Western"/"non-Western," differentiating also "intermediate" and "composite" lifestyles. In addition, recent programmatic studies have established guidelines for more ethical microbiome research (47, 51, 57, 65, 66). These guidelines emphasize the importance of involving Indigenous scientists and preventing bioprospecting while calling for

researchers to critically examine their own positionality within these power structures (6).

Even the influential journal *Nature* has published an editorial advocating for updating racial and ethnic categories in microbiome science (67). However, creating meaningful categories that accurately reflect human biological and cultural complexity requires more than refined scientific metrics alone. Addressing these categorizations in non-Western contexts necessarily requires the integration of social science methods and theoretical tools, particularly qualitative research approaches that can capture the subtleties of local contexts.

A call for a more interdisciplinary approach has already been launched, which implicates the call for integrating non-Western researchers or ontologies (6, 29, 47, 50, 68, 69). Productive collaborations between microbiologists and social scientists are needed not only in Global South contexts but equally in the Global North, where racial categories often remain unexamined and taken for granted. Interdisciplinary research (47), however, has warned against the risk to fall into the "subordinate-service mode of interdisciplinarity" (70), where a more critical gaze is brought in only in the phase of design or communication at an end stage of life science projects, without including new ontological and methodological logics. In conclusion, philosophers, historians, and anthropologists working in non-U.S. contexts could help understand not only how race is translated into different political and social outcomes outside the United States, but they could also question the ontological premises and historical backgrounds of U.S. racial categories, shedding light on different race constructs over the world.

## ACKNOWLEDGMENTS

This article is part of a project that has received funding from the European Union's Horizon 2020 Research and Innovation Programme ERC-HealthXCross (GA n. 949742).

## AUTHOR AFFILIATION

[1]Cà Foscari University of Venice, Venice, Veneto, Italy

## AUTHOR ORCIDs

Marta Scaglioni http://orcid.org/0000-0003-3019-9922

## FUNDING

| Funder | Grant(s) | Author(s) |
| --- | --- | --- |
| European Research Council | 949742 | Marta Scaglioni |

## ADDITIONAL FILES

The following material is available online.

### Open Peer Review

**PEER REVIEW HISTORY (review-history.pdf).** An accounting of the reviewer comments and feedback.

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
