## [Reviewer comments · mSystems]

Race by Other Names: Critiquing Population Categories in Global Microbiome Research

Marta Scaglioni

Corresponding Author(s): Marta Scaglioni, Universita Ca' Foscari Dipartimento di Filosofia e Beni Culturali

Review Timeline:

Submission Date:	February 19, 2025
Editorial Decision:	April 25, 2025
Revision Received:	May 21, 2025
Accepted:	June 26, 2025

Editor: Suzanne Ishaq

Reviewer(s): Disclosure of reviewer identity is with reference to reviewer comments included in decision letter(s). The following individuals involved in review of your submission have agreed to reveal their identity: Joseph L. Graves (Reviewer #1); abigail Nieves Delgado (Reviewer #2); Lianmin Chen (Reviewer #3)

Transaction Report:

DOI: <https://doi.org/10.1128/msystems.00216-25>

Re: mSystems00216-25 (**Beyond Population Categories: Decolonizing Human Difference in Global Microbiome Research**)

Dear Dr. Marta Scaglioni:

Revision Guidelines

Sincerely,
Suzanne Ishaq
Editor
mSystems

Reviewer #1 (Comments for the Author):

If your goal is to discuss decolonization, using the term "Global South" is not best. I would use the term "Neo-colonial" recognizing that these nations are still in a subservient relationship to their former colonizers. However, I do not consider this change in nomenclature as necessary to carry the paper's argument.

The authors need to define what they mean by biological race here. Just saying "race" or "racial" without a definition assumes that there is general agreement amongst biologists as to the meaning of this category. See for example discussion in: Graves JL. Favored Races in the Struggle for Life: Racism and the Speciation Concept. Cold Spring Harbor Perspectives Biol. 2023 Jul 18:a041454. doi: 10.1101/cshperspect.a041454; as well as in chapter two of Graves and Goodman, Racism, Not Race: Answers to Frequently Asked Questions (Columbia University Press), 2022.

At line 72 I also recommend citing: Also site PNAS USA report: National Academies of Sciences, Engineering, and Medicine; Division of Behavioral and Social Sciences and Education; Health and Medicine Division; Committee on Population; Board on Health Sciences Policy; Committee on the Use of Race, Ethnicity, and Ancestry as Population Descriptors in Genomics Research. Using Population Descriptors in Genetics and Genomics Research: A New Framework for an Evolving Field. Washington (DC): National Academies Press (US); 2023 Mar 14. PMID: 36989389.

Reviewer #2 (Comments for the Author):

The ms. "Beyond population categories: Decolonizing Human Difference in Global Microbiome Research" is a much-needed review of the state of the art of the debate of race in microbiome research. It presents a very complete overview of the critiques from philosophy and STS and a useful insight into the use of racializing categories in the field. I really want to see it published. That said, I think the author can revise some parts.

The title doesn't necessarily fit the content. The author could make clearer in which way the ms. Is contributing to decolonization, and perhaps what is meant with this.

Depending on the guidelines of the journal, the ms. could have a more general introduction. Starting with the Yanomami case feels rather specific and doesn't necessarily introduce the topic of the ms.

There is a paragraph that repeats itself in page 1 line 74 and in page 2 line 78 starting with "A third position..."

The explanation of what is going to be done in the ms. in page 2 lines 102 to 106 could give a bit more detail. The word "instead" in line 103 didn't work for me.

In page 5 line 169 the author introduces microbiome science as an example of the preceding paragraph which is on race in epigenetics. It is not clear to me why the author does it, and it is confusing. The author could include a transition here.

**Marta Scaglioni**
**Cà Foscari University of Venice**
**scaglioniemarta@gmail.com**

**Title**

**Beyond Population Categories: Decolonizing Human Difference in Global Microbiome**
**Research**

**Abstract**

This article examines how categories of human difference are conceptualized and deployed in
microbiome science, particularly in research conducted in Global South contexts. Through a
systematic review of 180 microbiome studies (2010-2024) and analysis of social science critiques, it
demonstrates how seemingly neutral terms like “population,” “lifestyle,” and “traditional” often
carry implicit racial assumptions that echo colonial scientific frameworks. While microbiome
science rarely uses explicit racial terminology, it frequently employs comparative frameworks that
create hierarchical distinctions between Western and non-Western populations. The article argues
that these categorizations stem not just from colonial legacies but from the comparative nature of
scientific knowledge production itself. It traces the evolution of racial thinking in science from post-
WWII genetics through contemporary microbiome research, showing how even as scientists reject
biological racial determinism, race persists as an analytical category through various proxies. It
concludes by examining recent efforts to develop more nuanced approaches to human
categorization in microbiome science and calls for deeper integration of social science perspectives
to avoid reproducing problematic categorizations while ensuring ethical and equitable research
practices.

**Keywords:**

microbiome science, race, populations, Global South, scientific categorization, comparisons

**Beyond Population Categories: Decolonizing Human Difference in Global** 38 **Microbiome Research**

**1. Introduction**

*1.1 The Yanomami as “Pristine” Research Objects*

Deep in the Venezuelan Amazon, a research team of microbiome scientists collects fecal samples
from the Yanomami, a population in the High Orinoco area of the Amazonas state in Venezuela,
describing them as an “uncontacted” population with “no documented previous contact with
Western people” (Clemente et al. 2015). They are studying microbes, whose analysis through
computational techniques has recently enabled researchers to understand their connection to human
health and disease. The scientists discover in these specimens “unprecedented levels of bacterial
diversity.” Their paper inserts itself into a pattern of studying Indigenous¹ and Global South
populations as pristine windows into humanity’s microbial past, carrying high levels of biodiversity
in light of their “uncontactedness” and “pre-modern” lifestyle. This narrative of isolation has come
under criticism by social sciences (Maroney 2017; Raffaetà 2022; De Lima Hutchison and Núñez
Casal 2023), since it positions these populations in an immutable past, overlooking decades of
change and contacts. The Yanomami, in fact, far from being “uncontacted,” have engaged with
researchers, missionaries, and government officials for generations.

*1.2 Race and population categories in scientific practice*

By positioning certain groups as “uncontacted,” microbiome research reproduces colonial patterns
of othering - creating subjects who exist outside of modern time and space (see Fabian 1983),
available for Western scientific extraction. This temporal and spatial distancing is enabled by the
use of taxonomies that echo earlier racial categories. While these studies often use seemingly
neutral categories such as “population,” “lifestyle,” “communities,” and “ethnic group,” these
categorizations often hide racial subtexts, especially when they are used to describe Global South
contexts (Benezra 2020). In this literature review, I examine how categories in microbiome research
function as pattern-based concepts connecting individuals to preconceived geographical or social
groupings. I focus specifically on microbiome research conducted in Global South contexts, where
these categorizations most clearly reveal their embedded Western assumptions and limitations.

While scientists generally reject biological racial determinism, debates persist about race’s utility as
an analytical category in research. Three distinct perspectives have emerged: firstly, mostly
geneticists argue that racial categories obscure rather than illuminate, emphasizing that human
genetic variation does not map onto socially defined racial groups (Barbujani 2006). Secondly,
epigenetic researchers often retain race as an analytical tool, using it as a proxy for socioeconomic
and demographic factors including healthcare access, financial stability, and social stress (King et
al. 2021). In this latter framework, race becomes operational – not as a biological reality, but as an
organizing principle to understand how social inequities become embodied and manifest in health
disparities (Yudell et al. 2016). A third position is held by clinicians who strongly believe in the
importance of race – as currently categorized – to monitor, understand, and intervene on differences
in health, but who reject the term and advocate for a replacement with “ethnicity” or “ethnic group,”

¹ Life sciences often use the term “Indigenous” as a broad substitute for “non-Westerners.” Unlike Global South populations, however, Indigenous peoples can live in the Global North.

considered to be less biology-based (Braveman and Parker Dominguez 2021). A third position is
held by clinicians who strongly believe in the importance of race – as currently categorized – to
monitor, understand, and intervene on differences in health, but who reject the term and advocate
for a replacement with “ethnicity” or “ethnic group,” considered to be less biology-based
(Braveman and Parker Dominguez 2021).

This article argues that the categorization observed in microbiome science, exemplified by the study
of the Yanomami, is highly asymmetrical as the Global North tends to define the Global South and
not vice versa. It argues that this dynamic must be understood through the lens of comparative
practices in scientific knowledge production. While Western populations receive nuanced,
contextualized classifications, non-Western groups are often reduced to broad, oversimplified
categories (De Lima Hutchison and Núñez Casal 2023). Categories used to define human groups in
microbiome science, as epistemic tools shaped by specific historical and cultural contexts, reflect
not just the mindset of individual scientists, but entire genealogies of Western scientific thought
about human difference: colonial scientific practices, post-WWII population genetics, and
contemporary biomedical frameworks. Their historical embeddedness in Western thought makes
them particularly problematic when applied to Global South contexts, where they often fail to
capture local social organizations, cultural practices, and demographic realities.

In the first section, I briefly trace the evolution of race categories in scientific research, from
genetics to contemporary microbiome science. The second section analyses how social scientists
have critiqued racial categorization in microbiome studies specifically. Finally, I examine how
microbiome research in the Global South employs seemingly neutral terms like “population” and
“communities,” and “lifestyle,” often pairing them with value-laden descriptors such as “rural” or
“traditional” that carry implicit racial assumptions. I conclude by tracing the social and life science
literature that offers programmatic suggestions for a more precise, equitable science.

*1.3 Methodological approach to literature review*

I will delineate the debates around the use of race categories in genetic and medical research
through secondary sources, both social and life science studies. The paragraph 3.2, instead, will be
based on an online search on PubMed, SCOPUS, and Web of Science, where I collected 180 papers
from 2010 to 2024 that deal specifically with human microbiome and the Global South (searched
also through the keywords “developing countries”).

**2. A Brief Genealogy of Race in Science: From WWII to Epigenetics**

*2.1 Post-war rejection of racial determinism*

[revised manuscript text omitted]

For example, in microbiome science, researchers have demonstrated strong correlations between
microbial diversity and social factors that are often racially patterned, such as socioeconomic status,
healthcare access, diet, and environmental exposures (Ishaq et al. 2019; 2021; Robinson et al. 2022;
Ishaq et al. 2022). The Human Microbiome Project (HMP),³ launched in 2007, is a groundbreaking
scientific initiative that focuses on studying the genomic composition of all microbes that live on
and in the human body, analyzing their collective DNA and investigating their role in human health
and disease. It considers racial backgrounds in associations with microbiome diversity and health-
related outcomes, concluding that the association is “strong” (Huttenhower et al. 2012). The racial
categories employed within the HMP – termed sometimes “race” and sometimes “ethnicities” – are
Asian, Black, Mexican, Puerto Rican, and White, clearly referring to a specific Western genetic
research tradition (Benezra 2020). Many microbiome studies in the US employ these same racial
categories (Chen et al. 2016; Renson et al. 2017; Sordillo et al. 2017; Carson et al. 2018), or refer to
African Americans as opposed to people of European ancestry (Fettweis et al. 2014), or Other
Nationalities as opposed to White Caucasians (Stearns et al. 2017).

In general, this reintegration of race into biomedical research carries similar risks. Critics from the
social sciences warn of a “molecular reinscription” of race (Duster 2006), that not only carries with
it a re-biologization of racial categories (Bliss 2012), but also racist narratives (Duster 2006; El-Haj
2007; Roberts 2011). In addition, while epigenetics aims to understand how social inequalities
become biologically embodied, its biosocial approach may inadvertently obscure crucial
socioeconomic factors affecting health outcomes (Chellappoo and Baedke 2023).

[revised manuscript text omitted]

This romanticized construction of rural environments as pristine habitats for “good” microbes has
faced substantial criticism (Maroney 2017; Benezra 2020; Nieves Delgado and Baedke 2021;
Raffaetà 2022). Such framing has enabled concerning attempts by Western researchers to
appropriate allegedly “missing” microbes from Global South populations (Benezra 2020; Rowland
2020; Nieves Delgado and Baedke 2021; Rawson 2024; Núñez Casal 2024), as exemplified by
microbiologist Jeff Leach’s widely-publicized fecal transplant involving Hadzabe (a Tanzanian
hunter-gatherer community) microbes (Leach 2014). Also, this perspective problematically frames
non-Western populations as “living fossils” (Maroney 2017) or “walking biobanks” (Raffaetà 2022,
47), collapsing distinctions between non-Western and pre-modern peoples. The assumption of
higher microbial diversity in non-Western contexts relies on a simplistic co-evolutionary framework
that treats contemporary non-Western individuals as proxies for ancestral humans with “ancient”
microbes (Mangola et al. 2022; Núñez Casal 2024). This framework suggests a straightforward
human-microbe co-evolution that has been increasingly questioned within microbiology itself
(Raffaetà 2022). Some critiques have emerged from within affected communities themselves -
notably, Hadzabe community member and human rights activist Shani Msafiri Mangola, who has
collaborated with microbiologists to challenge the romanticization of indigeneity (see also Nading
2016) as representing a lost connection to land and nature (Mangola et al. 2022). Furthermore,
microbiome research based on these racial assumptions often displays shallow contextual
knowledge (Benezra 2020; Rawson 2024) and overlooks local microbiota-impoverishing practices
(abuse of antibiotics, diffusion of formula milk) and high infection rates in the Global South
(Raffaetà 2022).

Recent programmatic studies have established guidelines for more ethical microbiome research
(Dominguez-Bello et al. 2016; Greenhough et al. 2020; Mangola et al. 2022; Bader et al. 2023;
Handsley-Davis et al. 2023). These guidelines emphasize the importance of involving Indigenous
scientists and preventing bioprospecting while calling for researchers to critically examine their
own positionality within these power structures (Benezra 2020).

[revised manuscript text omitted]

The asymmetry between Global North and Global South in microbiome research extends beyond
these conceptual categorizations. When examining the institutional affiliations and funding sources
of these comparative studies, a clear pattern emerges: 76.4% are led by Global North institutions,
with primary funding from Western sources (82.1%). Even when local researchers are involved, the
comparative frameworks typically originate from Western scientific paradigms. As South Africa-
based microbiologists Allali et al. (2021) demonstrate in their analysis of 168 microbiome research
papers across 33 African countries, most research papers on the African microbiome originate from
asymmetrical collaborations, where the first and/or last authors are affiliated with Western
universities, with major funders including the NIH, the Bill and Melinda Gates Foundation, and the
European Union.

While this systematic analysis reveals the persistence of problematic categorizations in microbiome
research, it also points toward opportunities to develop more nuanced, contextualized approaches
that move beyond rigid population categories and better reflect human biological and cultural
complexity.

4. Discussion: Beyond Population Categories. Recommendations for future research

Increasingly, in fact, microbiome scientists are questioning their use of categories to differentiate humans, their lifestyle, and the environment in which they live. For example, the study by biologists Sahana Kuthyar and Aspen T. Reese (2021) on the variations in microbial exposure at the human-animal interface problematizes dichotomies such as “Western”/“non-Western”, differentiating also “intermediate” and “composite” lifestyles. Their work suggests hybrid, non-conventional scenarios such as “Western industrial rural”; “non-Western, non-industrialized, or traditional rural”; “non-Western industrializing city”; and “Western industrialized city.”

Even the influential journal *Nature* has published an editorial advocating for updating racial and ethnic categories in microbiome science (‘Why Nature Is Updating Its Advice to Authors on Reporting Race or Ethnicity’ 2023). However, creating meaningful categories that accurately reflect human biological and cultural complexity requires more than refined scientific metrics alone. Addressing these categorizations in non-Western contexts necessarily requires the integration of social science methods and theoretical tools, particularly qualitative research approaches that can capture the subtleties of local contexts.

A call for a more interdisciplinary approach has already been launched, which implicates the call for integrating non-Western researchers or ontologies (Rees, Bosch, and Douglas 2018; Greenhough et al. 2020; Benezra 2020; Ishaq et al. 2021; De Wolfe et al. 2021; Robinson et al. 2022). Productive collaborations between microbiologists and social scientists are needed not only in Global South contexts but equally in the Global North, where racial categories often remain unexamined and taken for granted. Interdisciplinary research (Greenhough et al. 2020), however, has warned against the risk to fall into the “subordinate-service mode of interdisciplinarity” (Barry, Born, and Weszkalnys 2008), where a more critical gaze is brought in only in the phase of design or communication at an end stage of life science projects, without including new ontological and methodological logics. In conclusion, philosophers, historians and anthropologists working in non-US contexts could help understanding not only how race is translated into different political and social outcomes outside the US, but they could also question the ontological premises and historical backgrounds of US racial categories, shedding light on different race-constructs over the world.

References

- Albera, Dionigi. 2011. ‘Pour Un Compratisme Renouvelé’. In *Au Fil Des Générations. Terre, Pouvoir et Parenté Dans l’Europe Alpine (XIVe–XXe Siècles)*, Presses universitaires de Grenoble (PUG). Grenoble.
- Allali, Imane, Regina E. Abotsi, Lemese Ah. Tow, Lehana Thabane, Heather J. Zar, Nicola M. Mulder, and Mark P. Nicol. 2021. ‘Human Microbiota Research in Africa: A Systematic Review Reveals Gaps and Priorities for Future Research’. *Microbiome* 9 (1): 241. <https://doi.org/10.1186/s40168-021-01195-7>.
- Bader, Alyssa C., Essie M. Van Zuylen, Matilda Handsley-Davis, Rosanna A. Alegado, Amber Benezra, Rebecca M. Pollet, Hanareia Ehau-Taumaunu, Laura S. Weyrich, and Matthew Z. Anderson. 2023. ‘A Relational Framework for Microbiome Research with Indigenous Communities’. *Nature Microbiology* 8 (10): 1768–76. <https://doi.org/10.1038/s41564-023-01471-2>.
- Barbujani, Guido. 2006. *L’invenzione Delle Razze: Capire La Biodiversità Umana*. Tascabili Bompiani 353. Milano: Tascabili Bompiani.
- Barbujani, Guido, Arianna Magagni, Eric Minch, and L. Luca Cavalli-Sforza. 1997. ‘An Apportionment of Human DNA Diversity’. *Proceedings of the National Academy of Sciences* 94 (9): 4516–19. <https://doi.org/10.1073/pnas.94.9.4516>.
- Barry, Andrew, Georgina Born, and Gisa Weszkalnys. 2008. ‘Logics of Interdisciplinarity’. *Economy and Society* 37 (1): 20–49. <https://doi.org/10.1080/03085140701760841>.
- Benezra, Amber. 2020. ‘Race in the Microbiome’. *Science, Technology, & Human Values* 45 (5): 877–902. <https://doi.org/10.1177/0162243920911998>.

- Bliss, Catherine. 2012. *Race Decoded: The Genomic Fight for Social Justice*. Stanford University Press.
<https://doi.org/10.1515/9780804782050>.
- Boem, Federico. 2022. ‘Racism Afte the End of the Race: A Brief Epistemological Viewpoint on Genomic Studies and
Racism’. *Africa e Mediterraneo*, 2022.
- Bowker, G.C., and S.L. Star. 1999. *Sorting Things Out: Classification and Its Consequences*. Cambridge,
Massachussets and London: MIT Press. <https://books.google.it/books?id=Fon7ngEACAAJ>.
- Braveman, Paula, and Tyan Parker Dominguez. 2021. ‘Abandon “Race.” Focus on Racism’. *Frontiers in Public Health*
9:689462. <https://doi.org/10.3389/fpubh.2021.689462>.
- Brives, Charlotte, and Alexis Zimmer. 2021. ‘Ecologies and Promises of the Microbial Turn’. *Revue d’anthropologie*
*Des Connaissances* 15 (3). <https://doi.org/10.4000/rac.25068>.
- Candea, Matei. 2016. ‘De Deux Modalités de Comparaison En Anthropologie Sociale’. *L’Homme*, no. 218 (May), 183–
218. <https://doi.org/10.4000/lhomme.28968>.
- ———. 2019. *Comparison in Anthropology*. Cambridge University Press.
- Carson, Tiffany L., Fuchenchu Wang, Xiangqin Cui, Bradford E. Jackson, William J. Van Der Pol, Elliot J. Lefkowitz,
Casey Morrow, and Monica L. Baskin. 2018. ‘Associations between Race, Perceived Psychological Stress, and
the Gut Microbiota in a Sample of Generally Healthy Black and White Women: A Pilot Study on the Role of
Race and Perceived Psychological Stress’. *Psychosomatic Medicine* 80 (7): 640–48.
<https://doi.org/10.1097/PSY.0000000000000614>.
- Cavalli-Sforza, Luigi Luca, Paolo Menozzi, and Alberto Piazza. 1994. *The History and Geography of Human Genes*.
Princeton University Press.
- Chellappoo, Azita, and Jan Baedke. 2023. ‘Where the Social Meets the Biological: New Ontologies of Biosocial Race’.
*Synthese* 201 (1): 14. <https://doi.org/10.1007/s11229-022-04006-0>.
- Chen, Jun, Euijung Ryu, Matthew Hathcock, Karla Ballman, Nicholas Chia, Janet E. Olson, and Heidi Nelson. 2016.
‘Impact of Demographics on Human Gut Microbial Diversity in a US Midwest Population’. *PeerJ* 4:e1514.
<https://doi.org/10.7717/peerj.1514>.
- Clemente, Jose C., Erica C. Pehrsson, Martin J. Blaser, Kuldip Sandhu, Zhan Gao, Bin Wang, Magda Magris, et al.
2015. ‘The Microbiome of Uncontacted Amerindians’. *ScienceAdvances* 1 (e1500183).
<https://doi.org/10.1126/sciadv.1500183>.
- De Lima Hutchison, Coll, and Andrea Núñez Casal. 2023. ‘Sustaining (Dis)Embodied Inequalities in the(Ir) Eurocene:
Ancient Microbes, Racial Anthropometry, and Life Choices’. *Medicine Anthropology Theory* 10 (2): 1–33.
<https://doi.org/10.17157/mat.10.2.7105>.
- De Wolfe, Travis J., Mohammed Rafi Arefin, Amber Benezra, and María Rebolleda Gómez. 2021. ‘Chasing Ghosts:
Race, Racism, and the Future of Microbiome Research’. Edited by Kathryn C. Milligan-Myhre. *mSystems* 6
(5): e00604-21. <https://doi.org/10.1128/mSystems.00604-21>.
- Dominguez-Bello, Maria G., Daudi Peterson, Oscar Noya-Alarcon, Mariapia Bevilacqua, Nelson Rojas, Rómulo
Rodríguez, Saul Alango Pinto, Richard Baallow, and Hortensia Caballero-Arias. 2016. ‘Ethics of Exploring
the Microbiome of Native Peoples’. *Nature Microbiology* 1 (7): 16097.
<https://doi.org/10.1038/nmicrobiol.2016.97>.
- Duster, Troy. 2006. ‘The Molecular Reinscription of Race: Unanticipated Issues in Biotechnology and Forensic
Science’. *Patterns of Prejudice* 40 (4–5): 427–41. <https://doi.org/10.1080/00313220601020148>.
- El-Haj, Nadia Abu. 2007. ‘The Genetic Reinscription of Race’. *Annual Review of Anthropology* 36 (1): 283–300.
<https://doi.org/10.1146/annurev.anthro.34.081804.120522>.
- Fabian, J. 1983. *Time and the Other: How Anthropology Makes Its Object*. New York: Columbia University Press.
<https://books.google.it/books?id=6ZzVdQfAKVgC>.
- Fettweis, Jennifer M., J. Paul Brooks, Myrna G. Serrano, Nihar U. Sheth, Philippe H. Girerd, David J. Edwards, Jerome
F. Strauss, the Vaginal Microbiome Consortium, Kimberly K. Jefferson, and Gregory A. Buck. 2014.
‘Differences in Vaginal Microbiome in African American Women versus Women of European Ancestry’.
*Microbiology* 160 (10): 2272–82. <https://doi.org/10.1099/mic.0.081034-0>.
- Formosinho, Joana, Adam Bencard, and Louise Whiteley. 2022. ‘Environmentality in Biomedicine: Microbiome
Research and the Perspectival Body’. *Studies in History and Philosophy of Science* 91 (February):148–58.
<https://doi.org/10.1016/j.shpsa.2021.11.005>.
- Fujimura, Joan H., Troy Duster, and Ramya Rajagopalan. 2008. ‘Introduction: Race, Genetics, and Disease: Questions
of Evidence, Matters of Consequence’. *Social Studies of Science* 38 (5): 643–56.
<https://doi.org/10.1177/0306312708091926>.
- Gissis, Snait. 2008. ‘When Is “Race” a Race? 1946–2003’. *Studies in History and Philosophy of Science Part C:
Studies in History and Philosophy of Biological and Biomedical Sciences*, January.
https://www.academia.edu/54097359/When_is_race_a_race_1946_2003.
- Greenhough, Beth, Cressida Jervis Read, Jamie Lorimer, Javier Lezaun, Carmen McLeod, Amber Benezra, Sally
Bloomfield, et al. 2020. ‘Setting the Agenda for Social Science Research on the Human Microbiome’.
*Palgrave Communications* 6 (1): 18. <https://doi.org/10.1057/s41599-020-0388-5>.

Handsley-Davis, Matilda, Matthew Z. Anderson, Alyssa C. Bader, Hanareia Ehau-Taumaunu, Keolu Fox, Emma
Kowal, and Laura S. Weyrich. 2023. 'Microbiome Ownership for Indigenous Peoples'. *Nature Microbiology* 8
(10): 1777–86. <https://doi.org/10.1038/s41564-023-01470-3>.

Helmreich, Stefan. 2014. '*Homo Microbis*: The Human Microbiome, Figural, Literal, Political'. *Thresholds* 42
(January):52–59. https://doi.org/10.1162/thld_a_00076.

Huttenhower, Curtis, Dirk Gevers, Rob Knight, Sahar Abubucker, Jonathan H. Badger, Asif T. Chinwalla, Heather H.
Creasy, et al. 2012. 'Structure, Function and Diversity of the Healthy Human Microbiome'. *Nature* 486
(7402): 207–14. <https://doi.org/10.1038/nature11234>.

Ishaq, Suzanne L., Francisco J. Parada, Patricia G. Wolf, Carla Y. Bonilla, Megan A. Carney, Amber Benezra, Emily
Wissel, et al. 2021. 'Introducing the Microbes and Social Equity Working Group: Considering the Microbial
Components of Social, Environmental, and Health Justice'. *mSystems* 6 (4): e00471-21.
<https://doi.org/10.1128/mSystems.00471-21>.

Ishaq, Suzanne L., Maurisa Rapp, Risa Byerly, Loretta S. McClellan, Maya R. O'Boyle, Anika Nykanen, Patrick J.
Fuller, et al. 2019. 'Framing the Discussion of Microorganisms as a Facet of Social Equity in Human Health'.
*PLOS Biology* 17 (11): e3000536. <https://doi.org/10.1371/journal.pbio.3000536>.

Ishaq, Suzanne L., Emily F. Wissel, Patricia G. Wolf, Laura Grieneisen, Erin M. Eggleston, Gwynne Mhuireach,
Michael Friedman, et al. 2022. 'Designing the Microbes and Social Equity Symposium: A Novel
Interdisciplinary Virtual Research Conference Based on Achieving Group-Directed Outputs'. *Challenges* 13
(2): 30. <https://doi.org/10.3390/challe13020030>.

King, Dillon E., Pooja D. Lalwani, Gilberto Padilla Mercado, Emma L. Dolan, Johnna M. Frierson, Joel N. Meyer, and
Susan K. Murphy. 2024. 'The Use of Race Terms in Epigenetics Research: Considerations Moving Forward'.
*Frontiers in Genetics* 15 (January). <https://doi.org/10.3389/fgene.2024.1348855>.

Kowal, Emma. 2022. 'Structuring Race into the Machine: The Spoiled Promise of Postgenomic Sequencing
Technologies'. In *The Palgrave Handbook of the Anthropology of Technology*, edited by Maja Hojer Bruun,
Ayo Wahlberg, Rachel Douglas-Jones, Cathrine Hasse, Klaus Hoeyer, Dorthe Brogård Kristensen, and Brit
Ross Winthereik, 165–82. Singapore: Springer. https://doi.org/10.1007/978-981-16-7084-8_8.

Kuthyar, Sahana, and Aspen T. Reese. 2021. 'Variation in Microbial Exposure at the Human-Animal Interface and the
Implications for Microbiome-Mediated Health Outcome'. Edited by Suzanne L. Ishaq. *mSystems* 6 (4):
e00567-21. <https://doi.org/10.1128/mSystems.00567-21>.

Leach, J. 2014. 'Gut Microbiota: Please Pass the Microbes'. *Nature* 504 (7478): 33.

Lewin, Roger. 1993. 'Genes from a Disappearing World'. *New Scientist* (1971) 138 (1875): 25–29.

Lewontin, Richard C., Steven Peter Russell Rose, and Leon J. Kamin. 1984. *Not in Our Genes: Biology, Ideology, and*
*Human Nature*. Pantheon Books.

Lorimer, J. 2020. *The Probiotic Planet: Using Life to Manage Life*. Minneapolis: University of Minnesota Press.
<https://books.google.it/books?id=couKzQEACAAJ>.

Mangola, Shani Msafiri, Justin R. Lund, Stephanie L. Schnorr, and Alyssa N. Crittenden. 2022. 'Ethical Microbiome
Research with Indigenous Communities'. *Nature Microbiology* 7 (6): 749–56. <https://doi.org/10.1038/s41564-022-01116-w>.

Marks, Jonathan. 1995. 'The Human Genome Diversity Project: Good FOR If Not Good AS Anthropology'. 151568.
April 1995. <https://repository.library.georgetown.edu/handle/10822/527679>.

Maroney, Stephanie. 2017. 'Reviving Colonial Science in Ancestral Microbiome Research'. *MicrobioSocial* (blog).
2017. <https://microbiosocial.wordpress.com/2017/01/10/reviving-colonial-science-in-ancestral-microbiome-research/>.

M'charek, Aouatef Amâde. 2005. *The Human Genome Diversity Project: An Ethnography of Scientific Practice*.
Cambridge Studies in Society and the Life Sciences. Cambridge: Cambridge Univ. Press. http://bvbr.bib-bvb.de:8991/F?func=service&doc_library=BVB01&doc_number=013356876&line_number=0001&func_code=DB_RECORDS&service_type=MEDIA.

Nading, Alex. 2016. 'Evidentiary Symbiosis: On Paraethnography in Human–Microbe Relations'. *Science as Culture*
25 (4): 560–81. <https://doi.org/10.1080/09505431.2016.1202226>.

Nieves Delgado, Abigail, and Jan Baedke. 2021. 'Does the Human Microbiome Tell Us Something about Race?'
*Humanities and Social Sciences Communications* 8 (1): 1–12. <https://doi.org/10.1057/s41599-021-00772-3>.

———. 2024. 'How to Eliminate Race from Human Microbiome Research'. Università degli studi di Sassari.
<https://doi.org/10.14275/2465-2334/20240.NIE>.

Núñez Casal, Andrea. 2024. 'Race and Indigeneity in Human Microbiome Science: Microbiomisation and the
Historicity of Otherness'. *History and Philosophy of the Life Sciences* 46 (2): 17.
<https://doi.org/10.1007/s40656-024-00614-w>.

Paxson, H., and S. Helmreich. 2014. 'The Perils and Promises of Microbial Abundance: Novel Natures and Model
Ecosystems, from Artisanal Cheese to Alien Seas'. *Social Studies of Science* 44 (2): 165–93.
<https://doi.org/10.1177/0306312713505003>.

- Prainsack, Barbara. 2015. 'Is Personalized Medicine Different? (Reinscription: The Sequel) A Response to Troy Duster:
Is Personalized Medicine Different?' *The British Journal of Sociology* 66 (1): 28–35.
<https://doi.org/10.1111/1468-4446.12117>.
- Raffaetà, R. 2022. *Metagenomic Futures: How Microbiome Research Is Reconfiguring Health and What It Means to Be*
*Human*. Abingdon, Oxon ; New York, NY: Routledge.
- Rawson, Ariel Janaye. 2024. 'Anti-Racism, Racism, and the Microbiome: A Review'. *Progress in Environmental*
*Geography* 3 (2): 137–59. <https://doi.org/10.1177/27539687241245428>.
- Reardon, Jenny. 2009. *Race to the Finish: Identity and Governance in an Age of Genomics*. Princeton: Princeton
University Press. <https://muse.jhu.edu/pub/267/monograph/book/29887>.
- ———. 2017. *The Postgenomic Condition. Ethics, Justice & Knowledge After the Genome*. Chicago and London: The
University of Chicago Press.
- Rees, T., T. Bosch, and A.E. Douglas. 2018. 'How the Microbiome Challenges Our Concept of Self'. *PLoS Biol* 16 (2):
e2005358.
- Renson, Audrey, Heidi E. Jones, Francesco Beghini, Nicola Segata, Christine P. Zolnik, Mykhaylo Usyk, Thomas U.
Moody, et al. 2017. 'Sociodemographic Patterning in the Oral Microbiome of a Diverse Sample of New
Yorkers'. Preprint. *Epidemiology*. <https://doi.org/10.1101/189225>.
- Roberts, Dorothy E. 2011. *Fatal Invention: How Science, Politics, and Big Business Re-Crete Race in the Twenty-*
*First Century*. New York: New Press.
- Robinson, Jake M., Nicole Redvers, Araceli Camargo, Christina A. Bosch, Martin F. Breed, Lisa A. Brenner, Megan A.
Carney, et al. 2022. 'Twenty Important Research Questions in Microbial Exposure and Social Equity'.
*mSystems* 7 (1): e01240-21. <https://doi.org/10.1128/msystems.01240-21>.
- Romualdi, Chiara, David Balding, Ivane S. Nasidze, Gregory Risch, Myles Robichaux, Stephen T. Sherry, Mark
Stoneking, Mark A. Batzer, and Guido Barbujani. 2002. 'Patterns of Human Diversity, within and among
Continents, Inferred from Biallelic DNA Polymorphisms'. *Genome Research* 12 (4): 602–12.
<https://doi.org/10.1101/gr.214902>.
- Rosenberg, Noah A., Jonathan K. Pritchard, James L. Weber, Howard M. Cann, Kenneth K. Kidd, Lev A. Zhivotovsky,
and Marcus W. Feldman. 2002. 'Genetic Structure of Human Populations'. *Science* 298 (5602): 2381–85.
<https://doi.org/10.1126/science.1078311>.
- Rowland, Allison L. 2020. *Zoetropes and the Politics of Humanhood*. New Directions in Rhetoric and Materiality.
Columbus: The Ohio State University Press.
- Sellers, Christopher. 2018. 'To Place or Not to Place: Toward an Environmental History of Modern Medicine'. *Bulletin*
*of the History of Medicine* 92 (1): 1–45. <https://doi.org/10.1353/bhm.2018.0000>.
- Smits, Samuel A., Jeff Leach, Erica D. Sonnenburg, Carlos G. Gonzalez, Joshua S. Lichtman, Gregor Reid, Rob
Knight, et al. 2017. 'Seasonal Cycling in the Gut Microbiome of the Hadza Hunter-Gatherers of Tanzania'.
*Science* 357 (6353): 802–6. <https://doi.org/10.1126/science.aan4834>.
- Sordillo, Joanne E., Yanjiao Zhou, Michael J. McGeachie, John Ziniti, Nancy Lange, Nancy Laranjo, Jessica R. Savage,
et al. 2017. 'Factors Influencing the Infant Gut Microbiome at Age 3-6 Months: Findings from the Ethnically
Diverse Vitamin D Antenatal Asthma Reduction Trial (VDAART)'. *The Journal of Allergy and Clinical*
*Immunology* 139 (2): 482-491.e14. <https://doi.org/10.1016/j.jaci.2016.08.045>.
- Stearns, J. C., M. A. Zulyniak, R. J. de Souza, N. C. Campbell, M. Fontes, M. Shaikh, M. R. Sears, et al. 2017. 'Ethnic
and Diet-Related Differences in the Healthy Infant Microbiome.' *Genome Medicine* 9 (1).
<https://dx.doi.org/10.0.4.162/s13073-017-0421-5>.
- Stepan, Nancy. 1982. *Idea of Race in Science: Great Britain 1900-1960*. Archon Books.
- TallBear, Kimberly. 2013. *Native American DNA: Tribal Belonging and the False Promise of Genetic Science*.
Minneapolis, MN: University of Minnesota Press.
- 'Why Nature Is Updating Its Advice to Authors on Reporting Race or Ethnicity'. 2023. *Nature* 616 (7956): 219–219.
<https://doi.org/10.1038/d41586-023-00973-7>.

Dear Editor and Reviewers,

Thank you for your thorough and constructive feedback on my manuscript “Race by Other Names: Critiquing Population Categories in Microbiome Research.” I appreciate the time you have taken to review my work and offer valuable suggestions for improvement. I have carefully considered all your comments and revised the manuscript accordingly. Below, I outline the changes made in response to your feedback:

1. I have carefully verified and corrected the percentages in the systematic analysis of human microbiome articles, ensuring accurate representation of the correlations between population terms and comparative frameworks.
2. As suggested, I have incorporated Graves and Goodman (2023) when discussing the biological inconsistencies of race concepts. This addition has had to be accommodated to the word limits and therefore has not taken much space (highlighted).
3. Regarding the suggestion to replace “Global South” with “neo-colonial,” I understand the critique of this terminology. However, I have opted to keep “Global South” as it is widely employed in microbiome science literature and essential for my quantitative systematic term analysis. To acknowledge this, I have added a footnote explaining the contested nature of this terminology.
4. I appreciate the suggestion to include “Population Descriptors in Genetics and Genomics Research” by the NIH. I feel the manuscript needed an institutional perspective on racial categorization practices, complementing the scholarly work already cited.
5. Following your recommendation about the title, I have simplified it to “Race by Other Names: Critiquing Population Categories in Microbiome Research,” which reflects a more general content.
6. I have revised the introduction as suggested, removing the specific Yanomamo case study in favour of a more general introduction to race categories in microbiome science (highlighted). This includes noting how these categories are increasingly contested by official authorities, providing a stronger framing for the article.
7. I have eliminated the repetition you identified in the text.
8. While I was unable to expand on the programmatic articles regarding equitable microbiome science due to word limits, I have reorganized this section to include papers advocating for Indigenous involvement in research (highlighted in “Discussion”).
9. I have removed “instead” from line 103 as requested.
10. I have nuanced the framing of microbiome science as an example of race in epigenetic research throughout the article to address concerns about this characterization.

Thank you again for your valuable feedback that has helped strengthen this work.

Sincerely, Marta Scaglioni Cà Foscari University of Venice

Re: mSystems00216-25R1 (Race by Other Names: Critiquing Population Categories in Global Microbiome Research)

Dear Dr. Marta Scaglioni:

Your manuscript has been accepted, and I am forwarding it to the ASM production staff for publication. Your paper will first be checked to make sure all elements meet the technical requirements. ASM staff will contact you if anything needs to be revised before copyediting and production can begin. Otherwise, you will be notified when your proofs are ready to be viewed.

Sincerely,
Suzanne Ishaq
Editor
mSystems

Reviewer #1 (Comments for the Author):

The authors have successfully addressed my comments.

Reviewer #3 (Comments for the Author):

I think the current version reads well and could serve as an afternoon read for those working in microbiome population studies, while also encouraging caution in interpreting results observed between different populations.

Lianmin Chen
Nanjing Medical University